# Acute Effect of Caffeine-Based Multi-Ingredient Supplement on Reactive Agility and Jump Height in Recreational Handball Players

**DOI:** 10.3390/nu14081569

**Published:** 2022-04-09

**Authors:** Piotr Kaczka, Marcin Maciejczyk, Amit Batra, Anna Tabęcka-Łonczyńska, Marek Strzała

**Affiliations:** 1Professional Pharma, Body and Health Consulting, 35-106 Kielanówka, Poland; 2Department of Physiology and Biochemistry, University of Physical Education in Krakow, al. Jana Pawła II 78, 31-571 Kraków, Poland; marcin.maciejczyk@awf.krakow.pl; 3Batra Performance, 55-330 Lutynia, Poland; amit@op.pl; 4Department of Biotechnology and Cell Biology, Medical College, University of Information Technology and Management in Rzeszow, Sucharskiego 2, 35-225 Rzeszow, Poland; annaurz@wp.pl; 5Department of Water Sports, Faculty of Physical Education and Sport, University of Physical Education, 31-571 Kraków, Poland; marek.strzala@awf.krakow.pl

**Keywords:** agility test, countermovement jump, caffeine, multi-ingredient supplement

## Abstract

Pre-exercise caffeine and guarana-based multi-ingredient supplement (MS) consumption may be more effective for physical performance improvement than caffeine and guarana alone due to the synergistic effect of biologically active ingredients in multi-ingredient supplements. This study aimed to examine the acute effect of MS on the reactive agility and jump performance in recreational handball male players. A randomized, double-blind, crossover study involved twenty-four male handball players (body mass 74.6 ± 8.8 kg; body height 179 ± 7 cm; age 23.8 ± 1.4 years). Participants were tested under three conditions: placebo, caffeine + guarana (CAF + GUA), or MS ingestion 45 min before exercise tests. Participants performed a reactive agility test (Y-shaped test) and countermovement jump (CMJ). None of the supplements improved countermovement jump height (*p* = 0.06). The time needed to complete the agility test was significantly (*p* = 0.02) shorter in the MS condition than in the placebo. The differences in agility between PL vs. CAF + GUA and MS vs. CAF + GUA conditions were not statistically significant (*p* = 0.88 and *p* = 0.07, respectively). The results of this study indicate that the caffeine-based multi-ingredient performance was effective in improvement in reactive agility but not in jump height in recreational handball male players. A similar effect was not observed with CAF + GUA ingestion alone.

## 1. Introduction

Muscle power and agility are strongly associated with athletic performance. Agility and power are needed in such sports as tennis and badminton, Taekwon-Do, or team sports [1]. Sheppard and Young [2] defined reactive agility as a rapid whole-body movement with a change in velocity or direction in response to a stimulus. In many sports disciplines, agility and/or speed changes can result in a score or a shift in the game’s momentum [3]. On the other hand, muscle power is essential to perform dynamic movements requiring both speed and strength. It is necessary for speed and power sports and those where jumping ability is needed, such as team games such as handball.

Many athletes believe that pre-workout supplementation improves concentration, decreases reaction time, increases power and endurance, and reduces fatigue [4,5]. The most popular pre-workout supplement is caffeine (CAF), which enhances performance through peripheral and central mechanisms [6,7,8,9]. The effects of CAF ingestion on aerobic performance are well documented [10], and previous studies also focused on the impact of CAF consumption on anaerobic performance [11,12]. However, the effects of CAF on muscle strength, power, speed, and agility are inconsistent [12]. Previous studies reported a slight improvement in reactive agility [13] or no effect [10] of CAF at the same dose of 6 mg/kg. Regarding jump height (power), similarly, both improvement [14] and no effect [15] of caffeine were observed. The type of exercise test used, participant characteristics (e.g., age and training experience), and form of CAF (or composition of multi-ingredient supplement) may be responsible for the inconsistent results [11].

The previous study [16] reported an increase in jump height after caffeinated drink consumption but, at the same time, emphasized that it was unknown whether the observed effect was due to caffeine content or the presence of other substances such as taurine. After caffeine-containing energy drinks consumption, an increase in strength and endurance and jumping performance was observed, but the improvement in physical performance was associated with taurine dosage [17]. Pre-exercise multi-ingredient pre-workout supplements are a specialized category of dietary supplements that include a blend of ingredients aimed to increase exercise performance [18,19]. Many components that are used in MS affect different physiological mechanisms. It is supposed that some combinations of these ingredients may increase the effectiveness of a given preparation in improving physical performance [4,20]. Ingredients found in many MS, such as CAF, arginine, beta-alanine, creatine, citrulline, or different plant ingredients, may also have synergistic effects on improving athletic performance [4,20,21,22]. Caffeine and other sympathomimetic ingredients of MS could be more effective in performance improvement than CAF per se. The dopaminergic and catecholaminergic impact may also be enhanced by tyrosine, a component of many MS. Multiple ingredients potentially interact, increasing or decreasing supplement effectiveness. Pre-workout MS typically consists of numerous active ingredients, which can modify pharmacodynamics and pharmacokinetics and thus result in different bioavailability properties and physiological effects, such as in the case of various amino acids [23,24].

It was hypothesized that ingesting MS, containing both CAF and guarana (GUA), may be more effective than consuming CAF + GUA alone in improving agility and power—the synergistic effect of biologically active ingredients in multi-ingredient supplements based on caffeine and guarana may be more effective than CAF + GUA ingestion per se. Therefore, this study aimed to examine the acute effects of an MS on reactive agility and jump height in recreationally-trained handball male players.

Given the importance of jumping ability and agility in many popular sports, determining the effect of supplementation with caffeine-based supplements on agility and power would be of great scientific and practical interest.

## 2. Materials andMethods

### 2.1. Study Design

Players were asked to refrain from consuming caffeine, tea, any additional supplements or ergogenic aids, and alcohol-containing products for the two weeks preceding the survey and not doing strenuous training for 24 h before the examination. The participants were asked not to consume anything for 3 h before the test and not to change their diet during the study. Participants were asked to report any side effects after ingestion of the given supplements.

All participants attended a familiarization session one week before the study. During the familiarization session, participants were instructed on how to perform the test and performed two trials for each test after a prior individual warm-up. In the study’s main part, the players were tested under three conditions: placebo, caffeine and guarana (CAF + GUA), and MS supplementation in random order. Participants repeated the testing session every three days. Exercise tests were performed around noon (10.00–13.00). After consuming either a placebo or the supplement solution (CAF + GUA or MS), handball players rested for 15 min. Standard warm-up started with 10 min run at 60–75% of maximal heart rate (heart rate was self-controlled during warm-up). Maximal heart rate was calculated using the 211–0.64·age formula [25]. After the run, participants performed various dynamic stretching exercises for five minutes. First, the agility test was performed, followed by a countermovement jump (CMJ).

### 2.2. Participants

Twenty–four recreationally-trained handball male players (body mass = 74.6 ± 8.8 kg; body height = 179 ± 7 cm; age = 23.8 ± 1.4 years) were involved in this study. Participants trained up to three pieces of training per week with medium to high-intensity (including resistance training) and played one match (academic league) in a week. Players were recruited from academic handball teams according to the criteria for inclusion and exclusion. Inclusion criteria were as follow: men, age 18–45, good general health determined based on a medical examination, including the assessment of resting ECG and resting blood pressure. Exclusion criteria were hypersensitivity to any of the product components (verified based on a declaration in a personal questionnaire) and no injury in the six months preceding the study. Players were acquainted with the purpose and course of research. They also provided their written consent to participate in the project. The study protocol was approved by the Ethical Committee of the University of Physical Education in Katowice (Poland; opinion No. 2/2018) and conformed to the ethical requirements of the 1975 Helsinki Declaration.

### 2.3. Supplementation

Forty-five minutes before testing, players were randomly provided with either (a) placebo (PL): 250 mL of the flavored water; or (b) caffeine, flavored water containing: anhydrous CAF (200 mg) (Biesterfeld International, Poland) and guarana extract (200 mg) (EVER Pharma, Lyon, France): 300 mg of CAF in total, mixed with water (250 mL) (CAF + GUA condition); or (c) 9.6 g MS powder, (Olimp, Poland), mixed with 250 mL of water. MS contained: L-citrulline (3 g), beta-alanine (2 g), taurine (750 mg), L-arginine (500 mg), L-tyrosine, anhydrous caffeine (200 mg), guarana extract (200 mg; in total 300 mg of caffeine), barley extract (150 mg), cayenne pepper seed extract (25 mg), black pepper extract (7.5 mg) and Huperzia Serrata extract (3 mg). The guarana extract was standardized for 50% caffeine content. The amount of CAF consumed by players was the same in the CAF + GUA and MS conditions, and its dosage was about 5 mg/kg.

### 2.4. Reactive Agility Test: 1-1-2 (Y-Shaped Test)

Four pairs of electronic timing gates systems (Fusion Smart Speed PRO, Brisbane, Australia) were set, as pointed out in Figure 1. Participants began each trial 20 cm in front of the starting line. After the first 5 m run at maximum speed in a straight line toward the second timing gate, the system indicates the further direction of the subject’s movement. After crossing the middle gate, the next gate lights turned on, forcing the participant to change direction as quickly as possible while maintaining the maximum possible speed run for the last 10 m. Participants performed two trials. They completed a left or right run in random order after running the first 5 m. The better time from these two trials was used for analysis. The test 1-1-2 scheme is shown in Figure 1.

### 2.5. Countermovement Jump

The Optojump system (Optojump, Microgate, Bolzano, Italy) consisting of two bars (transmitting and receiving, 1 m apart) was used to evaluate the jump height in the countermovement jump. CMJ was performed without arm swing (i.e., hands placed on hips). Participants were instructed to start from an erect position and make a downward movement before taking off the floor. During the CMJ, there was no interval for rest between the two phases of the jump (eccentric and concentric phases). All players performed two trials with a 60 s interval between each attempt. The better result of these two trials was taken for further analysis as previously described [26,27,28].

### 2.6. Statistical Analysis

The data are presented as means and standard deviations (mean ± SD). The significance of differences between conditions was performed using analysis of variance (ANOVA) with repeated measures. Post hoc analysis was carried out using Tukey’s test. Data distribution was checked using the Shapiro–Wilk test. Homogeneity of variance within the groups was tested via Levene’s test (variance of the analyzed parameters was similar in both groups). The effect size (partial eta squared (η^2^) was calculated and interpreted as small (0.01), medium (0.06), or large (0.14) [29]. Additionally, in post hoc analysis, the effect size (d-Cohen: ES) between conditions was calculated and interpreted as small (0.20), medium (0.50), or large (0.80) [29]. Statistically significant results were defined as a *p*-value of <0.05. The following software was used to perform the calculations: STATISTICA 13.1 (StatSoft, Tulsa, OK, USA).

## 3. Results

No side effects were observed. The reactive agility results observed a statistically significant difference between the three conditions (f = 4.24, *p* = 0.02). The time needed to complete the agility test was significantly shorter in MS conditions than in PL (−3.4%; *p* = 0.02). The differences in agility between PL vs. CAF + GUA and MS vs. CAF + GUA conditions were not statistically significant (*p* = 0.88 and *p* = 0.07, respectively). There was no significant difference between jump height conditions (f = 2.89, *p* = 0.06) (Table 1).

## 4. Discussion

This study aimed to examine the acute effects of an MS on reactive agility and jump height in recreationally-trained handball male players. Our study showed that ingestion of MS before exercise improved the players’ agility but did not improve jumping height. Surprisingly, a similar effect was not observed after CAF + GUA ingestion alone—ingestion of CAF + GUA alone had no significant impact on either agility or jumping power. The observed effects of MS ingestion were not a result of CAF + GUA alone but rather a synergistic effect of other active substances contained in MS. The reported *p*-values of 0.06 (CMJ, ANOVA) and 0.07 (CAF + GUA-MS) in the post hoc analysis (agility) suggest a trend for a likely effect. To our best knowledge, this is the first study comparing MS and the same amount of CAF on reactive agility and jump height.

Comparing various MS or MS vs. a single ingredient is difficult and often impossible due to the frequent use of “proprietary blends” that do not disclose specific ingredient amounts. However, it was indicated that pre-exercise ingestion of MS may positively affect endurance, although ambiguous data were reported on the acute impact of MIPS on strength and power [20]. Studies to date on the effectiveness of MS consumption are inconclusive [20]. Conflicting results were reported regarding the effect of MS on upper-body and lower-body power production [20]. A previous paper reported both improvement [4] and lack of progress [30] in sprint performance. MS ingestion has little effect on jumping performance [20]. Our results align with data presented by Lane and Byrd [31], who found no effect of acute MS ingestion on vertical jump performance. Spradley et al. [30] reported that MS ingestion might benefit reaction time in recreationally-trained males.

Training status (trained vs. untrained) appears to play an essential role in response to caffeine intake [11]. Improved reactive agility may be limited to trained individuals as no ergogenic effect may be observed in untrained individuals performing the pro agility test [10]. Moreover, the result tends to be stronger for exercises involving large muscle groups [32]. Participants in this study were physically active. However, they were not advanced or elite players, yet we noted improved agility after MS ingestion, but not after caffeine consumption. Our results are in line with Rocha et al. [15], who indicated that pre-exercise CAF ingestion is not effective in improving the upper and lower limb muscle power in handball players. Grgic et al. [33] suggested that ingesting a placebo or CAF may enhance CMJ performance. Still, no significant effects of condition were found on maximal power output generated during takeoff. In contrast, the meta-analysis by Grgic et al. [11] supports CAF as an effective for increases in muscle power expressed as vertical jump height. Another study conducted on a group of elite volleyball players also showed improvements in CMJ parameters such as flight time (+5.3%), peak power (+16.2%), and peak concentric force (+6.5%) without any side effects of CAF ingestion in the amount of 5 mg/kg [34].

Caffeine affects cognitive performance in a dose-dependent and person-specific manner, especially in the case of sleep deprivation or comparing the effect in tired versus well-rested people [35,36]. Pre-exercise CAF ingestion between 3.0 and 6.0 mg/kg seems to be a safe ergogenic aid for players in team sports. However, the efficacy of caffeine varies depending on various factors such as the nature of the game, physical status, and caffeine habituation [37]. However, on the other hand, “participants” habituation status with CAF does not seem to affect either aerobic or anaerobic exercise [6]. The previous studies indicated the positive effect of CAF in the dose range of 32–300 mg (0.5–4.0 mg/kg) on the central nervous system and essential cognitive functions enhancing arousal and the ability to concentrate [38] and attention, vigilance, and reaction time [35,39,40]. However, the limitation of the studies based on pre-planned tests was that none tested the caffeine’s effect on reactive agility in-game conditions and demanding behaviors during the match when a perceptual component forces players to initiate the movement response [37]. In this study, a dose of 5 mg/kg was used, which may have been too low for the players’ training status. Pontifex [41] indicated that even higher doses of CAF (6 mg/kg body mass) had no influence on reactive agility time measured with a modality similar to this study.

In our study, participants consumed the same amount of CAF in both conditions (MS and CAF + GUA), but the effects were different. Although we did not report a result of both conditions on jump height, MS was more effective than CAF + GUA on agility. This confirms our hypothesis that this may be a synergistic effect of the substances in MS. The impact of CAF could be enhanced by other ingredients which share an additive (synergistic) or similar mechanism of action. Considering that plant alkaloids such as piperine or capsaicin, present in the examined MS, may cause increased secretion of catecholamines [42]. In addition to caffeine, guarana seeds also contain potentially psychoactive components such as flavonoids, saponins, and tannins, positively stimulating cognitive functions [43,44].

Our data could be helpful for strength and conditioning specialists and suggest that MS could be helpful and effective in enhancing agility and perhaps (trend: *p* = 0.06) jumping power in recreational handball players. A limitation of this study is the training status of the participants and the sport they practice. The effects may differ in other athletes with varying statuses of training and other sports disciplines. Multi-ingredient supplements with a different composition than those used in this study may induce different effects.

## 5. Conclusions

Pre-exercise ingestion of the caffeine-based MS can significantly improve reactive agility performance but not jump height in recreational handball male players. Further comparative studies (MS ingestion vs. only CAF + GUA or CAF ingestion) are needed—a similar effect was not observed with CAF + GUA ingestion alone. It also seems necessary to determine which combination of MS components will have the most significant impact on improving performance.

## Figures and Tables

**Figure 1 nutrients-14-01569-f001:**
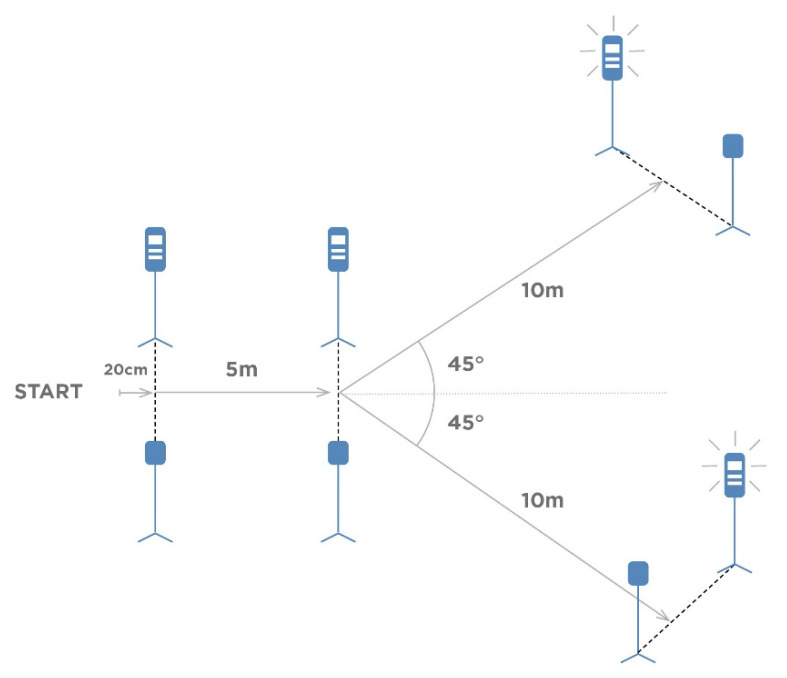
The layout of the photocell timing gates and the way of carrying out the test task protocol 1-1-2.

**Table 1 nutrients-14-01569-t001:** Effects of supplementation on reactive agility test and countermovement jump.

TEST	PL	MS	CAF + GUA	ANOVA *p* (f)	η^2^	POST-HOC *p* [ES]
Y-shaped [s]	2.595 ± 0.142	2.507 ± 0.087	2.58 ± 0.102	0.02 (4.24)	0.07	PL-MS:0.02 [0.77] CAF + GUA-MS: 0.07 [0.77] PL-CAF: 0.88 [0.12]
CMJ [cm]	48.12 ± 4.09	50.61 ± 4.01	48.16 ± 4.19	0.06 (2.89)	0.11	

PL—placebo; MS—multi-ingredient supplement; CAF + GUA—caffeine and guarana; CMJ—countermovement jump, 1-1-2—agility test, ES—effect size.

## Data Availability

The data presented in this study are available on request from the corresponding author.

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
