# Peer review of "Acute Effect of Caffeine-Based Multi-Ingredient Supplement on Reactive Agility and Jump Height in Recreational Handball Players"

_nutrients, 2022, doi:10.3390/nu14081569_

Round 1
Reviewer 1 Report
This study aimed to to examine the acute effect of a caffeine-based multi-ingredient supplement (MS) on the reactive agility and jump performance in recreational handball male players. Since the investigation could be useful for Strength and Conditioning specialists, there are several edits and concerns I need to resolve:
Abstract.
Delete last sentence “Further comparative studies (MS ingestion vs. only caffeine ingestion) and MS with different compositions are needed.”
Material and methods.
It is really important, necessary and essential that handball players carry out a period of familiarization with the tests. Explain if it was so.
Move these sentences to participants section, please:
“Players were recruited from academic handball teams according to the criteria for inclusion and exclusion. Inclusion criteria were as follow: men, age 18-45, good general health determined based on a medical examination including the assessment of resting ECG and resting blood pressure. Exclusion criteria were hypersensitivity to any of the product components (verified based on a declaration in a personal questionnaire) and no injury in the six months preceding the study”
“Players were acquainted with the purpose and course of research. They also provided their written consent to participate in the project. In the case of soccer players. The study was approved by the Ethical Committee of the University of Physical Education in Katowice (Poland; opinion No. 2/2018) and conformed to the ethical requirements of the 1975 Helsinki Declaration”.
I think that the performing test every three days is not correct. Players should carry out testing sessions at the same day of the week (i.e., microcycle).
Use abbreviation CAF since the first citation in-text.
Use abbreviation CMJ since the first citation in-text.
Include more descriptive variables of participants. For example, IMC and %Fat.
Reactive agility test 1-1-2. Was this test validated in previous investigation? Include ICC, CV… Why is this test relevant in a handball context? Could you ensure this test correlates with the actions in handball matches?
Close parentheses after “concentric phases”.
Which test of ANOVA. I think you should perform repeated-measures ANOVA.
In my opinion discussion section should be completely restructured as well as include some important data from the cited studies. I propose the following classification of paragraphs.
- Aims, main results, novelty (same as current version).
- Comparisons of results in Reactive Agility Test: 1-1-2 (Y-test). How affect consuming CAF?
- Comparisons of results in CMJ. How affect consuming CAF?
- (include that this data is limited to specific sample).
In two central paragraphs authors should include sport modality and competitive-level of participants for each investigation. Also, postulate practical applications.
References.
Please, unify format.
Only two references has been published in the last two years. Please, include new references. Revise some interesting systematic review of consuming caffeine.
Author Response
Reviewer 1
We would like to thank you for evaluating our work and providing us with helpful comments allowing us to improve the manuscript. Below, we present our point-by-point responses to the Reviewer's comments.
This study aimed to examine the acute effect of a caffeine-based multi-ingredient supplement (MS) on the reactive agility and jump performance in recreational handball male players. Since the investigation could be useful for Strength and Conditioning specialists, there are several edits and concerns I need to resolve:
Abstract.
Delete last sentence "Further comparative studies (MS ingestion vs. only caffeine ingestion) and MS with different compositions are needed."
Response: Deleted as suggested
Material and methods.
It is really important, necessary and essential that handball players carry out a period of familiarization with the tests. Explain if it was so.
Response: We added the description of the familiarization session to the text as suggested.
Move these sentences to participants section, please:
"Players were recruited from academic handball teams according to the criteria for inclusion and exclusion. Inclusion criteria were as follow: men, age 18-45, good general health determined based on a medical examination including the assessment of resting ECG and resting blood pressure. Exclusion criteria were hypersensitivity to any of the product components (verified based on a declaration in a personal questionnaire) and no injury in the six months preceding the study"
"Players were acquainted with the purpose and course of research. They also provided their written consent to participate in the project. In the case of soccer players. The study was approved by the Ethical Committee of the University of Physical Education in Katowice (Poland; opinion No. 2/2018) and conformed to the ethical requirements of the 1975 Helsinki Declaration".
Response: Done.
I think that the performing test every three days is not correct. Players should carry out testing sessions at the same day of the week (i.e., microcycle).
Response: You're right. In professional athletes, it would have a significant impact. However, note that our players were training handball recreationally, thus not in the same training rigor as elite or advanced players. In our opinion, this was not as significant for our subjects as it was for the advanced players.
Use abbreviation CAF since the first citation in-text.
Response: corrected
Use abbreviation CMJ since the first citation in-text.
Response: corrected
Include more descriptive variables of participants. For example, IMC and %Fat.
Response: Unfortunately, we only performed basic somatic measurements (BH and BM) in the players.
Reactive agility test 1-1-2. Was this test validated in previous investigation? Include ICC, CV… Why is this test relevant in a handball context? Could you ensure this test correlates with the actions in handball matches?
Response: A Y-shaped test (1-1-2), as a reactive agility test was used in this study. This assessment is common used in team sport and is reliable and valid. For this reason, we did not revalidate this test in this study.
References:
Oliver, J.L.; Meyers, R.W. Reliability and generality of measures of acceleration, planned agility, and reactive agility. Int. J. Sports Physiol. Perform. 2009, 4, 345–354.
Spasic, M., Krolo, A., Zenic, N., Delextrat, A., & Sekulic, D. (2015). Reactive agility performance in handball; development and evaluation of a sport-specific measurement protocol. Journal of sports science & medicine, 14(3), 501.
Sheppard, J. M., & Young, W. B. (2006). Agility literature review: Classifications, training and testing. Journal of sports sciences, 24(9), 919-932.
Paul, D. J., Gabbett, T. J., & Nassis, G. P. (2016). Agility in team sports: Testing, training and factors affecting performance. Sports Medicine, 46(3), 421-442.
Close parentheses after "concentric phases".
Response: Done.
Which test of ANOVA. I think you should perform repeated-measures ANOVA.
Response: Yes, it was ANOVA with repeated measure. Added.
In my opinion discussion section should be completely restructured as well as include some important data from the cited studies. I propose the following classification of paragraphs.
- Aims, main results, novelty (same as current version).
- Comparisons of results in Reactive Agility Test: 1-1-2 (Y-test). How affect consuming CAF?
- Comparisons of results in CMJ. How affect consuming CAF?
- (include that this data is limited to specific sample).
In two central paragraphs authors should include sport modality and competitive-level of participants for each investigation. Also, postulate practical applications.
Response: Practical application was added. We focused the discussion on comparing the effects with other data and possible reasons for these differences, such as training status or amount of caffeine consumed. We have added a new paragraph on the MS's reported effects on power, jump and sprint performance, and reaction time. To our knowledge, there are few articles (all referenced in this paper) on the effects of MS supplementation on agility and jump performance. In the discussion, we also wanted to avoid comparing the impact of different MS in different groups or with other exercise tests with our data. We also do not want to speculate about physiological mechanisms responsible for the observed effects - we did not investigate them in this study.
References. Please, unify format. Only two references has been published in the last two years. Please, include new references. Revise some interesting systematic review of consuming caffeine.
Response: We added some new references (10) and unified the format.
Reviewer 2 Report
Thank you for allowing me to read this interesting manuscript. Overall, it is clearly presented. Other than some minor points, detailed below, I do feel that the authors need to strengthen the rationale for the inclusion of a multi-ingredient supplement; is this because it is frequently used by athletes, or might to be more beneficial to performance than caffeine alone.
Abstract
It would be useful to add a brief, one sentence, rationale for the study in the background section.
Please include participant characteristics
Support your description of the results with some data and associated statistics.
Introduction
Overall, the introduction provides a strong rationale for the methods employed, but it is not so clear why a multi-ingredient supplement may be better, or at least as effective as caffeine alone. What are the potential benefits of ingesting a multi-ingredient supplement that athletes would not obtain from caffeine?
Methods
When highlighting that ‘the participants were asked not to consume anything for 3 hours before the test’, it would be helpful to state what time testing took place. Was the diet otherwise standardised?
Results
The authors may wish to consider including the effect sizes for the post-hoc analysis
Discussion
This does read a little like a literature review. It would be useful if the mechanisms were more explicitly related to the present study. Furthermore, what are the key practical applications of these findings. Do these results suggest that a multi-ingredient supplement is not require, and caffeine alone can produce the same effects? Why does CMJ not improve with caffeine / multi-ingredient supplement; is this a different mechanism?
I hope that the authors find the above comments helpful and in the constructive manner they are intended.
Author Response
Reviewer 2
Thank you for your contribution to improving our paper. We appreciate your work. We have made the necessary changes. Please, find our responses to your comments.
Thank you for allowing me to read this interesting manuscript. Overall, it is clearly presented. Other than some minor points, detailed below, I do feel that the authors need to strengthen the rationale for the inclusion of a multi-ingredient supplement; is this because it is frequently used by athletes, or might to be more beneficial to performance than caffeine alone.
Response: We have strengthened the introduction by adding a paragraph about MS.
Abstract
It would be useful to add a brief, one sentence, rationale for the study in the background section.
Response: Added as suggested
Please include participant characteristics
Response: Added.
Support your description of the results with some data and associated statistics.
Response: Added.
Introduction
Overall, the introduction provides a strong rationale for the methods employed, but it is not so clear why a multi-ingredient supplement may be better, or at least as effective as caffeine alone. What are the potential benefits of ingesting a multi-ingredient supplement that athletes would not obtain from caffeine?
Response: We have strengthened the introduction by adding a paragraph about MS.
Methods
When highlighting that 'the participants were asked not to consume anything for 3 hours before the test', it would be helpful to state what time testing took place. Was the diet otherwise standardised?
Response: Testing was performed around midday, and testing hours (10.00-13.00) were similar for all players. Participants were also asked not to change their diet during the study. Added to the manuscript.
Results
The authors may wish to consider including the effect sizes for the post-hoc analysis
Response: Included as suggested – added to the table and statistical method section
Discussion
This does read a little like a literature review. It would be useful if the mechanisms were more explicitly related to the present study. Furthermore, what are the key practical applications of these findings. Do these results suggest that a multi-ingredient supplement is not require, and caffeine alone can produce the same effects? Why does CMJ not improve with caffeine / multi-ingredient supplement; is this a different mechanism?
Response: In the discussion and the introduction, we avoided describing possible mechanisms of the ergogenic effects of caffeine - we did not study possible mechanisms in our subjects, we do not know what the mechanism of the observed effects was, and we do not want to speculate. In the introduction, we hinted that the mechanism of caffeine action might be peripheral or central without giving details. We focused the discussion on comparing the effects with other data and possible reasons for these differences such as training status or amount of caffeine consumed. We added information about practical applications of the results at the end of the discussion
I hope that the authors find the above comments helpful and in the constructive manner they are intended.
Response: Thank you for your helpful comments, which have significantly improved our paper.
Reviewer 3 Report
Please use no headings in the abstract.
I suggest to be more specific in the final statement of the abstract. In addition, why would you suggest MS ingestion vs only caffeine? Please clarify.
Ref 3 does not provide observations from a study to support the statement.
Ref 6 is dated. Please provide a more recent source that caffeine is the most popular pre-workout supplement.
“in recent years” in the introduction but Ref 9 is almost 10 years old. Please revise.
“in previous studies” but then only one reference is provided. Please change.
“The previous study [13] reported an increase in jump height after caffeine ingestion”. It was a caffeinated drink. Please clarify.
“A significant relationship was shown between taurine content and performance but not between caf-feine content and performance [14].” Please be spedific, performance is too general.
“including the assessment of resting ECG and resting blood pressure.”. Pleaser provide the observations.
What is “strong tea”. Please clarify.
“attended a familiarization session one week”. What was done in the familiarizaiton session? Please clarify.
“60%–75% of maximal heart rate”. How was maximal heart rate determined? In addition, please provide the heart rate values for the various conditions.
Please delete “In the case of soccer players”
I suggest to change “body height = 179±7.2 cm” to “body height = 179±7 cm”
One of the conditions was guarana and caffeine. Please clarify throughout the manuscript.
Please provide details for the “chemical form of caffeine”.
What is meant by “The best trial for each side was used for analysis”. Please clarify.
“No side effects were observed”. There is no mention of the recording of side effects. How was that measured/recorded? Please clarify.
Time is expressed with three decimal places. Was the time measured with three decimal places by the electronic timing gates
P-values of 0.06 and 0.07 seem to suggest that the study was underpowered. Sometimes, such -values are taken as a trend for a likely effect. Please reconsider.
What time of the day was the testing. Was this the same for each participant? Please clarify.
According to description of the Y-test (Participants had completed the test until two sprints to either side were recorded.), so participants sprinted 40 m. Are the times correct?
“for the fitness level of the study group.”. What was the fitness level. Any physiological parameter?
I suggest to provide realistic limitations. The MS has so many components that it would be logistically challenging to examine the limitation and I cannot see other examining that limitation.
Also, the fact that chronic intake was not examined is not really a limitation but was just not in the aim of the study. Please revise.
Please complete required info for Institutional Review Board Statement etc.
Author Response
Reviewer 3
We would like to thank you for evaluating our work and providing us with your helpful comments that allowed to improve our manuscript. Below, we present our point-by-point responses to the Reviewer's comments.
Please use no headings in the abstract.
Response: corrected
I suggest to be more specific in the final statement of the abstract. In addition, why would you suggest MS ingestion vs only caffeine? Please clarify.
Response: The last sentence is deleted as Reviewer 1 suggested.
Ref 3 does not provide observations from a study to support the statement.
Response: Agreed, but this statement is the citation from this paper (first introduction sentence). In the original, 'In many sports, changes of speed or rapid and decisive changes of direction can result in a break, a score or a shift in the momentum of the game.'
Ref 6 is dated. Please provide a more recent source that caffeine is the most popular pre-workout supplement.
Response: We added some new references.
"in recent years" in the introduction but Ref 9 is almost 10 years old. Please revise.
Response: Revised.
"in previous studies" but then only one reference is provided. Please change.
Response: Revised.
"The previous study [13] reported an increase in jump height after caffeine ingestion". It was a caffeinated drink. Please clarify.
Response: Revised.
"A significant relationship was shown between taurine content and performance but not between caf-feine content and performance [14]." Please be spedific, performance is too general.
Response: This sentence is revised
"including the assessment of resting ECG and resting blood pressure.". Pleaser provide the observations..
Response: Unfortunately, we did not record this data. These examinations were only performed for medical qualification for participation in the project to exclude possible contraindications to involvement.
What is "strong tea". Please clarify.
Response: Thank you for this comment. It is indeed difficult to define this phrase. Corrected
"attended a familiarization session one week". What was done in the familiarizaiton session? Please clarify.
Response: During the familiarization session, participants were briefed on how to perform the test and performed two trials for each test after a prior individual warm-up. Added to the manuscript.
"60%–75% of maximal heart rate". How was maximal heart rate determined? In addition, please provide the heart rate values for the various conditions.
Response: HRmax was calculated using the 211 − 0.64·age formula (Nes et al., 2013). Then, each participant's target HR was estimated to be between 60-76%HRmax and during the warm-up player self-controlled his heart rate to stay within the assumed range. We did not record the actual heart rate from the warm-up.
Reference:
Nes, B. M., Janszky, I., Wisløff, U., Støylen, A., & Karlsen, T. (2013). Age‐predicted maximal heart rate in healthy subjects: The HUNT Fitness Study. Scandinavian journal of medicine & science in sports, 23(6), 697-704.
Please delete "In the case of soccer players"
Response: Thank you for this copyediting mistake! Deleted!
I suggest to change "body height = 179±7.2 cm" to "body height = 179±7 cm"
Response: Done.
One of the conditions was guarana and caffeine. Please clarify throughout the manuscript.
Response: Thank you for this suggestion. Clarified. We introduce new abbreviation CAF+GUA throughout the manuscript.
Please provide details for the "chemical form of caffeine".
Response: Chemical form i.e. anhydrous caffeine and guarana extract. Described in methods section so we revised this sentence for clarification.
What is meant by "The best trial for each side was used for analysis". Please clarify.
Response: Participants performed two trials. They completed a left or right run in random order after running the first 5 m. The better time from these two trials was used for analysis. Revised.
"No side effects were observed". There is no mention of the recording of side effects. How was that measured/recorded? Please clarify.
Response: Participants were asked to report any side effects after ingesting the given supplements. Under all conditions, none of them reported any side effects.
Time is expressed with three decimal places. Was the time measured with three decimal places by the electronic timing gates
Response: Yes, time was measured with three decimal places.
P-values of 0.06 and 0.07 seem to suggest that the study was underpowered. Sometimes, such -values are taken as a trend for a likely effect. Please reconsider.
Response: Thank you for this suggestion. We added this interpretation to the first paragraph of the discussion.
What time of the day was the testing. Was this the same for each participant? Please clarify.
Response: Testing was performed around midday, and testing hours (10.00-13.00) were similar for all players. Added to manuscript.
According to description of the Y-test (Participants had completed the test until two sprints to either side were recorded.), so participants sprinted 40 m. Are the times correct?
The subjects covered 15 meters (5m+10m) in one trial (fig.1.). The time presented is for one trial (subjects performed two trials), it is correct.
"for the fitness level of the study group.". What was the fitness level. Any physiological parameter?
Response: Thank you for your comment. We did not determine the fitness level of the participants in this study - in this sentence we meant, as we wrote earlier in the discussion, the training status of the players. We have corrected this sentence to be more precise.
I suggest to provide realistic limitations. The MS has so many components that it would be logistically challenging to examine the limitation and I cannot see other examining that limitation. Also, the fact that chronic intake was not examined is not really a limitation but was just not in the aim of the study. Please revise.
Response: You are right, it would be challenging to evaluate. The realistic limitation of this study is the characteristics of participants and their physical performance (training status). The effects may differ in other athletes with different training statuses and other sports disciplines. Multi-ingredient supplements with a different composition than those used in this study may induce different effects. We have reordered this section by also removing the limitation about chronic effects.
Please complete required info for Institutional Review Board Statement etc.
Response: Completed.
Round 2
Reviewer 1 Report
Authors have addressed all my concerns.
Author Response
Dear Mr. Reviewer,
Thank you for your in-depth review and pertinent comments, which we will undoubtedly pay attention to in the future.
Yours faithfully
Piotr Kaczka
Reviewer 3 Report
The introduction still has information that indicates that a multi-supplement will be examined vs caffeine alone, but that is not the case as it is caffeine and guarana. For example, "It was hypothesized that ingesting MS may be more effective than consuming caffeine alone...." Please revise.
Change "CAFcaffeine" in the discussion.
Author Response
Dear Reviewer 3,
we apologize for this omission. Thank you again for your insightful review and helpful suggestions.